# OFF-DYNAMICS REINFORCEMENT LEARNING:
## TRAINING FOR TRANSFER WITH DOMAIN CLASSIFIERS

**Benjamin Eysenbach**[*]
CMU, Google Brain
beysenba@cs.cmu.edu

**Shreyas Chaudhari**[*]
CMU
shreyaschaudhari@cmu.edu

**Swapnil Asawa**[*]
University of Pittsburgh
swa12@pitt.edu

**Sergey Levine**
UC Berkeley, Google Brain

**Ruslan Salakhutinov**
CMU

## ABSTRACT

We propose a simple, practical, and intuitive approach for domain adaptation in reinforcement learning. Our approach stems from the idea that the agent's experience in the source domain should look similar to its experience in the target domain. Building off of a probabilistic view of RL, we achieve this goal by compensating for the difference in dynamics by modifying the reward function. This modified reward function is simple to estimate by learning auxiliary classifiers that distinguish source-domain transitions from target-domain transitions. Intuitively, the agent is penalized for transitions that would indicate that the agent is interacting with the source domain, rather than the target domain. Formally, we prove that applying our method in the source domain is guaranteed to obtain a near-optimal policy for the target domain, provided that the source and target domains satisfy a lightweight assumption. Our approach is applicable to domains with continuous states and actions and does not require learning an explicit model of the dynamics. On discrete and continuous control tasks, we illustrate the mechanics of our approach and demonstrate its scalability to high-dimensional tasks.

## 1 INTRODUCTION

Reinforcement learning (RL) can automate the acquisition of complex behavioral policies through real-world trial-and-error experimentation. However, many domains where we would like to learn policies are not amenable to such trial-and-error learning, because the errors are too costly: from autonomous driving to flying airplanes to devising medical treatment plans, safety-critical RL problems necessitate some type of *transfer learning*, where a safer source domain, such as a simulator, is used to train a policy that can then function effectively in a target domain. In this paper, we examine a specific transfer learning scenario that we call domain adaptation, by analogy to domain adaptation problems in computer vision (Csurka, 2017), where the training process in a source domain can be modified so that the resulting policy is effective in a given target domain.

RL algorithms today require a large amount of experience in the *target domain*. However, for many tasks we may have access to a different but structurally similar *source domain*. While the source domain has different dynamics than the target domain, experience in the source domain is much cheaper to collect. However, transferring policies from one domain to another is challenging because strategies which are effective in the source domain may not be effective in the target domain. For example, aggressive driving may work well on a dry racetrack but fail catastrophically on an icy road.

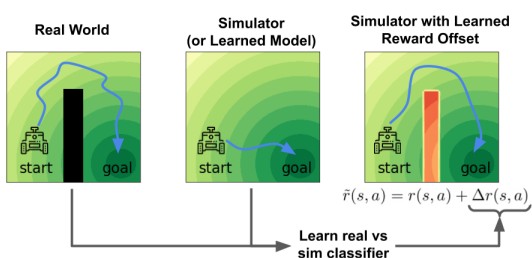

Figure 1: Our method acquires a policy for the target domain by practicing in the source domain using a (learned) modified reward function.

---

[*]Equal contribution.

While prior work has studied the domain adaptation of *observations* in RL (Bousmalis et al., 2018; Ganin et al., 2016; Higgins et al., 2017), it ignores the domain adaptation of the *dynamics*.

This paper presents a simple approach for domain adaptation in RL, illustrated in Fig. 1. Our main idea is that the agent's experience in the source domain should look similar to its experience in the target domain. Building off of a probabilistic view of RL, we formally show that we can achieve this goal by *compensating for the difference in dynamics by modifying the reward function*. This modified reward function is simple to estimate by learning auxiliary classifiers that distinguish source-domain transitions from target-domain transitions. Because our method learns a classifier, rather than a dynamics model, we expect it to handle high-dimensional tasks better than model-based methods, a conjecture supported by experiments on the 111-dimensional Ant task. Unlike prior work based on similar intuition (Koos et al., 2012; Wulfmeier et al., 2017b), a key contribution of our work is a formal guarantee that our method yields a near-optimal policy for the target domain.

The main contribution of this work is an algorithm for domain adaptation to dynamics changes in RL, based on the idea of compensating for differences in dynamics by modifying the reward function. We call this algorithm Domain Adaptation with Rewards from Classifiers, or DARC for short. DARC does not estimate transition probabilities, but rather modifies the reward function using a pair of classifiers. We formally analyze the conditions under which our method produces near-optimal policies for the target domain. On a range of discrete and continuous control tasks, we both illustrate the mechanics of our approach and demonstrate its scalability to higher-dimensional tasks.

## 2 RELATED WORK

While our work will focus on domain adaptation applied to RL, we start by reviewing more general ideas in domain adaptation, and defer to Kouw & Loog (2019) for a recent review of the field. Two common approaches to domain adaptation are importance weighting and domain-agnostic features. *Importance-weighting methods* (e.g., (Zadrozny, 2004; Cortes & Mohri, 2014; Lipton et al., 2018)) estimate the likelihood ratio of examples under the target domain versus the source domain, and use this ratio to re-weight examples sampled from the source domain. Similar to prior work on importance weighting (Bickel et al., 2007; Sønderby et al., 2016; Mohamed & Lakshminarayanan, 2016; Uehara et al., 2016), our method will use a classifier to estimate a probability ratio. Since we will need to estimate the density ratio of conditional distributions (transition probabilities), we will learn two classifiers. Importantly, we will use the logarithm of the density ratio to modify the reward function instead of weighting samples by the density ratio, which is often numerically unstable (see, e.g., Schulman et al. (2017, §3)) and led to poor performance in our experiments.

Prior methods for applying domain adaptation to RL include approaches based on system identification, domain randomization, and observation adaptation. Perhaps the most established approach, system identification (Ljung, 1999), uses observed data to tune the parameters of a simulator (Feldbaum, 1960; Werbos, 1989; Wittenmark, 1995; Ross & Bagnell, 2012; Tan et al., 2016; Zhu et al., 2017b; Farchy et al., 2013) More recent work has successfully used this strategy to bridge the sim2real gap (Chebotar et al., 2019; Rajeswaran et al., 2016). Closely related is work on online system identification and meta-learning, which directly uses the inferred system parameters to update the policy (Yu et al., 2017; Clavera et al., 2018; Tanaskovic et al., 2013; Sastry & Isidori, 1989). However, these approaches typically require either a model of the environment or a manually-specified distribution over potential test-time dynamics, requirements that our method will lift. Another approach, *domain randomization*, randomly samples the parameters of the source domain and then finds the best policy for this randomized environment (Sadeghi & Levine, 2016; Tobin et al., 2017; Peng et al., 2018; Cutler et al., 2014). While often effective, this method is sensitive to the choice of which parameters are randomized, and the distributions from which these simulator parameters are sampled. A third approach, *observation adaptation*, modifies the observations of the source domain to appear similar to those in the target domain (Fernando et al., 2013; Hoffman et al., 2016; Wulfmeier et al., 2017a). While this approach has been successfully applied to video games (Gamrian & Goldberg, 2018) and robot manipulation (Bousmalis et al., 2018), it ignores the fact that the source and target domains may have differing dynamics.

Finally, our work is similar to prior work on transfer learning (Taylor & Stone, 2009) and meta-learning in RL, but makes less strict assumptions than most prior work. For example, most work on meta-RL (Killian et al., 2017; Duan et al., 2016; Mishra et al., 2017; Rakelly et al., 2019) and

some work on transfer learning (Perkins et al., 1999; Tanaka & Yamamura, 2003; Sunmola & Wyatt, 2006) assume that the agent has access to many source tasks, all drawn from the same distribution as the target task. Selfridge et al. (1985); Madden & Howley (2004) assume a manually-specified curriculum of tasks, Ravindran & Barto (2004) assume that the source and target domains have the same dynamics locally, and Sherstov & Stone (2005) assume that the set of actions that are useful in the source domain is the same as the set of actions that will be useful in the target domain. Our method does not require these assumptions, allowing it to successfully learn in settings where these prior works would fail. For example, the assumption of Sherstov & Stone (2005) is violated in our experiments with broken robots: actions which move a joint are useful in the source domain (where the robot is fully-function) but not useful in the target domain (where that joint is disabled). Our method will significantly outperform an importance weighting baseline (Lazaric, 2008). Unlike Vemula et al. (2020), our method does not require learning a dynamics model and is applicable to stochastic environments and those with continuous states and actions. Our algorithm bears a resemblance to that in Wulfmeier et al. (2017b), but a crucial algorithmic difference allows us to prove that our method acquires a near-optimal policy in the target domain, and also leads to improved performance empirically.

The theoretical derivation of our method is inspired by prior work which formulates control as a problem of probabilistic inference (e.g., (Toussaint, 2009; Rawlik et al., 2013; Levine et al., 2018)). Algorithms for model-based RL (e.g., (Deisenroth & Rasmussen, 2011; Hafner et al., 2018; Janner et al., 2019)) and off-policy RL (e.g., (Munos et al., 2016; Fujimoto et al., 2018; Dann et al., 2014; Dudík et al., 2011) similarly aim to improve the sample efficiency of RL, but do use the source domain to accelerate learning. Our method is applicable to any maximum entropy RL algorithm, including on-policy (Song et al., 2019), off-policy (Abdolmaleki et al., 2018; Haarnoja et al., 2018), and model-based (Janner et al., 2019; Williams et al., 2015) algorithms. We will use the SAC (Haarnoja et al., 2018) in our experiments and compare against model-based baselines.

## 3 PRELIMINARIES

In this section, we introduce notation and formally define domain adaptation for RL. Our problem setting will consider two MDPs: $\mathcal{M}_{\text{source}}$ represents the source domain (e.g., a practice facility, simulator, or learned approximate model of the target domain) while $\mathcal{M}_{\text{target}}$ represents a the target domain. We assume that the two domains have the same state space $\mathcal{S}$, action space $\mathcal{A}$, reward function $r$, and initially state distribution $p_1(s_1)$; the only difference between the domains is the dynamics, $p_{\text{source}}(s_{t+1} \mid s_t, a_t)$ and $p_{\text{target}}(s_{t+1} \mid s_t, a_t)$. We will learn a Markovian policy $\pi_\theta(a \mid s)$, parametrized by $\theta$. Our objective is to learn a policy $\pi$ that maximizes the expected discounted sum of rewards on $\mathcal{M}_{\text{target}}$, $\mathbb{E}_{\pi, \mathcal{M}_{\text{target}}}[\sum_t \gamma^t r(s_t, a_t)]$. We now formally define our problem setting:

**Definition 1.** *Domain Adaptation for RL is the problem of using interactions in the source MDP $\mathcal{M}_{source}$ together with a small number of interactions in the target MDP $\mathcal{M}_{target}$ to acquire a policy that achieves high reward in the target MDP, $\mathcal{M}_{target}$.*

We will assume every transition with non-zero probability in the target domain will have non-zero probability in the source domain:

$$p_{\text{target}}(s_{t+1} \mid s_t, a_t) > 0 \implies p_{\text{source}}(s_{t+1} \mid s_t, a_t) > 0 \qquad \text{for all } s_t, s_{t+1} \in \mathcal{S}, a_t \in \mathcal{A}. \quad (1)$$

This assumption is common in work on importance sampling (Koller & Friedman, 2009, §12.2.2), and the converse need not hold: transitions that are possible in the source domain need not be possible in the target domain. If this assumption did not hold, then the optimal policy for the target domain might involve behaviors that are not possible in the source domain, so it is unclear how one could learn a near-optimal policy by practicing in the source domain.

## 4 A VARIATIONAL PERSPECTIVE ON DOMAIN ADAPTATION IN RL

The probabilistic inference interpretation of RL (Kappen, 2005; Todorov, 2007; Toussaint, 2009; Ziebart, 2010; Rawlik et al., 2013; Levine, 2018) treats the reward function as defining a desired distribution over trajectories. The agent's task is to sample from this distribution by picking trajectories with probability proportional to their exponentiated reward. This section will reinterpret this model in the context of domain transfer, showing that domain adaptation of *dynamics* can be done

by modifying the *rewards*. To apply this model to domain adaptation, define $p(\tau)$ as the desired distribution over trajectories in the target domain,

$$p(\tau) \propto p_1(s_1)\left( \prod_t p_{\text{target}}(s_{t+1} \mid s_t, a_t) \right) \exp\left( \sum_t r(s_t, a_t) \right),$$

and $q(\tau)$ as our agent's distribution over trajectories in the source domain,

$$q(\tau) = p_1(s_1) \prod_t p_{\text{source}}(s_{t+1} \mid s_t, a_t)\pi_\theta(a_t \mid s_t).$$

As noted in Section 3, we assume both trajectory distributions have the same initial state distribution. Our aim is to learn a policy whose behavior in the source domain both receives high reward and has high likelihood under the target domain dynamics. We codify this objective by minimizing the reverse KL divergence between these two distributions:

$$\min_{\pi(a|s)} D_{\text{KL}}(q \parallel p) = -\mathbb{E}_{p_{\text{source}}}\left[ \sum_t r(s_t, a_t) + \mathcal{H}_\pi[a_t \mid s_t] + \Delta r(s_{t+1}, s_t, a_t) \right] + c, \quad (2)$$

where

$$\Delta r(s_{t+1}, s_t, a_t) \triangleq \log p(s_{t+1} \mid s_t, a_t) - \log q(s_{t+1} \mid s_t, a_t).$$

The constant $c$ is the partition function of $p(\tau)$, which is independent of the policy and dynamics. While $\Delta r$ is defined in terms of transition probabilities, in Sec. 5 we show how to estimate $\Delta r$ by learning a classifier. We therefore call our method *domain adaptation with rewards from classifiers* (DARC), and will use $\pi^*_{\text{DARC}}$ to refer to the policy that maximizes the objective in Eq. 2.

Where the source and target dynamics are equal, the correction term $\Delta r$ is zero and we recover maximum entropy RL (Ziebart, 2010; Todorov, 2007). The reward correction is different from prior work that adds $\log \beta(a \mid s)$ to the reward to regularize the policy to be close to the behavior policy $\beta$ (e.g., Jaques et al. (2017); Abdolmaleki et al. (2018)). In the case where the source dynamics are *not* equal to the true dynamics, this objective is not the same as maximum entropy RL on trajectories sampled from the source domain. Instead, this objective suggests a corrective term $\Delta r$ that should be added to the reward function to account for the discrepancy between the source and target dynamics. The correction term, $\Delta r$, is quite intuitive. If a transition $(s_t, a_t, s_{t+1})$ has equal probability in the source and target domains, then $\Delta r(s_t, a_t) = 0$ so no correction is applied. For transitions that are likely in the source but are unlikely in the target domain, $\Delta r < 0$, the agent is penalized for "exploiting" inaccuracies or discrepancies in the source domain by taking these transitions. For the example environment in Figure 1, transitions through the center of the environment are blocked in the target domain but not in the source domain. For these transitions, $\Delta r$ would serve as a large penalty, discouraging the agent from taking these transitions and instead learning to navigate around the wall. Appendix A presents additional interpretations of $\Delta r$ in terms of coding theory, mutual information, and a constraint on the discrepancy between the source and target dynamics. Appendix C discusses how prior work on domain agnostic feature learning can be viewed as a special case of our framework.

## 4.1 THEORETICAL GUARANTEES

We now analyze when maximizing the modified reward $r + \Delta r$ in the source domain yields a near-optimal policy for the target domain. Our proof relies on the following lightweight assumption:

**Assumption 1.** *Let* $\pi^* = \arg\max_\pi \mathbb{E}_p\left[\sum r(s_t, a_t)\right]$ *be the reward-maximizing policy in the target domain. Then the expected reward in the source and target domains differs by at most* $2R_{max}\sqrt{\epsilon/2}$:

$$\left| \mathbb{E}_{p_{\pi^*, source}}\left[ \sum r(s_t, a_t) \right] - \mathbb{E}_{\pi^*, p_{target}}\left[ \sum r(s_t, a_t) \right] \right| \leq 2R_{max}\sqrt{\epsilon/2}.$$

The variable $R_{\text{max}}$ refers to the maximum entropy-regularized return of any trajectory. This assumption says that the optimal policy in the target domain is still a good policy for the source domain, and its expected reward is similar in both domains. We do not require that the opposite be true: the optimal policy in the source domain does not need to receive high reward in the target domain. If there are multiple optimal policies, we only require that this assumption hold for one of them. We now state our main result:

**Algorithm 1** Domain Adaptation with Rewards from Classifiers [DARC]

1: **Input:** source MDP $\mathcal{M}_{\text{source}}$ and target $\mathcal{M}_{\text{target}}$; ratio $r$ of experience from source vs. target.
2: **Initialize:** replay buffers for source and target transitions, $\mathcal{D}_{\text{source}}, \mathcal{D}_{\text{target}}$; policy $\pi$; parameters $\theta = (\theta_{\text{SAS}}, \theta_{\text{SA}})$ for classifiers $q_{\theta_{\text{SAS}}}(\text{target} \mid s_t, a_t, s_{t+1})$ and $q_{\theta_{\text{SAS}}}(\text{target} \mid s_t, a_t)$.
3: **for** $t = 1, \cdots$, num iterations **do**
4:     $\mathcal{D}_{\text{source}} \leftarrow \mathcal{D}_{\text{source}} \cup \text{ROLLOUT}(\pi, \mathcal{M}_{\text{source}})$             ▷ Collect source data.
5:     **if** $t \mod r = 0$ **then**               ▷ Periodically, collect target data.
6:         $\mathcal{D}_{\text{target}} \leftarrow \mathcal{D}_{\text{target}} \cup \text{ROLLOUT}(\pi, \mathcal{M}_{\text{target}})$
7:     $\theta \leftarrow \theta - \eta \nabla_\theta \ell(\theta)$               ▷ Update both classifiers.
8:     $\tilde{r}(s_t, a_t, s_{t+1}) \leftarrow r(s_t, a_t) + \Delta r(s_t, a_t, s_{t+1})$     ▷ $\Delta r$ is computed with Eq. 3.
9:     $\pi \leftarrow \text{MAXENT RL}(\pi, \mathcal{D}_{\text{source}}, \tilde{r})$
10: **return** $\pi$

**Theorem 4.1.** *Let $\pi^*_{DARC}$ be the policy that maximizes the modified (entropy-regularized) reward in the source domain, let $\pi^*$ be the policy that maximizes the (unmodified, entropy-regularized) reward in the target domain, and assume that $\pi^*$ satisfies Assumption 1. Then the following holds:*

$$\mathbb{E}_{p_{target}, \pi^*_{DARC}} \left[ \sum r(s_t, a_t) + \mathcal{H}[a_t \mid s_t] \right] \geq \mathbb{E}_{p_{target}, \pi^*} \left[ \sum r(s_t, a_t) + \mathcal{H}[a_t \mid s_t] \right] - 4R_{max}\sqrt{\epsilon/2}.$$

See Appendix B for the proof and definition of $\epsilon$. This result says that $\pi^*_{\text{DARC}}$ attains near-optimal (entropy-regularized) reward on the target domain. Thus, we can expect that modifying the reward function should allow us to adapt to different dynamics. The next section will present a practical algorithm for acquiring $\pi^*_{\text{DARC}}$ by estimating and effectively maximizing the modified reward in the source domain.

## 5    DOMAIN ADAPTATION IN RL WITH A LEARNED REWARD

The variational perspective on model-based RL in the previous section suggests that we should modify the reward in the source domain by adding $\Delta r$. In this section we develop a practical algorithm for off-dynamics RL by showing how $\Delta r$ can be estimated without learning an explicit dynamics model.

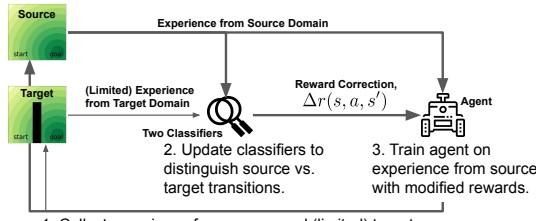

To estimate $\Delta r$, we will use a pair of (learned) binary classifiers, which will infer whether transitions came from the source or target domain. The key idea is that the transition probabilities are related to the classifier probabilities via Bayes' rule:

Figure 2: Block diagram of DARC (Alg. 1)

$$p(\text{target} \mid s_t, a_t, s_{t+1}) = \underbrace{p(s_{t+1} \mid s_t, a_t, \text{target})}_{=p_{\text{target}}(s_{t+1} \mid s_t, a_t)} p(s_t, a_t \mid \text{target})p(\text{target})/p(s_t, a_t, s_{t+1}).$$

We estimate the term $p(s_t, a_t \mid \text{target})$ on the RHS via *another* classifier, $p(\text{target} \mid s_t, a_t)$:

$$p(s_t, a_t \mid \text{target}) = \frac{p(\text{target} \mid s_t, a_t)p(s_t, a_t)}{p(\text{target})}.$$

Substituting these expression into our definition for $\Delta r$ and simplifying, we obtain an estimate for $\Delta r$ that depends solely on the predictions of these two classifiers:

$$\Delta r(s_t, a_t, s_{t+1}) = \log p(\text{target} \mid s_t, a_t, s_{t+1}) - \log p(\text{target} \mid s_t, a_t)$$
$$- \log p(\text{source} \mid s_t, a_t, s_{t+1}) + \log p(\text{source} \mid s_t, a_t) \tag{3}$$

The orange terms are the difference in logits from the classifier conditioned on $s_t, a_t, s_{t+1}$, while the blue terms are the difference in logits from the classifier conditioned on just $s_t, a_t$. Intuitively,

$\Delta r$ answers the following question: for the task of predicting whether a transition came from the source or target domain, how much better can you perform after observing $s_{t+1}$? We make this connection precise in Appendix A.2 by relating $\Delta r$ to mutual information. Ablation experiments (Fig. 7) confirm that both classifiers are important to the success of our method. The use of transition classifiers makes our method look somewhat similar to adversarial imitation learning (Ho & Ermon, 2016; Fu et al., 2017). While our method is *not* solving an imitation learning problem (we do not assume access to any expert experience), our method can be interpreted as learning a *policy* such that the dynamics observed by that policy in the source domain imitate the dynamics of the target domain.

**Algorithm Summary**   Our algorithm modifies an existing MaxEnt RL algorithm to additionally learn two classifiers, $q_{\theta_{\text{SAS}}}(\text{target} \mid s_t, a_t, s_{t+1})$ and $q_{\theta_{\text{SA}}}(\text{target} \mid s_t, a_t)$, parametrized by $\theta_{\text{SAS}}$ and $\theta_{\text{SA}}$ respectively, to minimize the standard cross-entropy loss.

$$\ell_{\text{SAS}}(\theta_{\text{SAS}}) \triangleq -\mathbb{E}_{\mathcal{D}_{\text{target}}}\left[\log q_{\theta_{\text{SAS}}}(\text{target} \mid s_t, a_t, s_{t+1})\right] - \mathbb{E}_{\mathcal{D}_{\text{source}}}\left[\log q_{\theta_{\text{SA}}}(\text{source} \mid s_t, a_t, s_{t+1})\right]$$

$$\ell_{\text{SA}}(\theta_{\text{SA}}) \triangleq -\mathbb{E}_{\mathcal{D}_{\text{target}}}\left[\log q_{\theta_{\text{SA}}}(\text{target} \mid s_t, a_t)\right] - \mathbb{E}_{\mathcal{D}_{\text{source}}}\left[\log q_{\theta_{\text{SA}}}(\text{source} \mid s_t, a_t)\right].$$

Our algorithm, Domain Adaptation with Rewards from Classifiers (DARC), is presented in Alg. 1 and illustrated in Fig. 2. To simplify notation, we define $\theta \triangleq (\theta_{\text{SAS}}, \theta_{\text{SA}})$ and $\ell(\theta) \triangleq \ell_{\text{SAS}}(\theta_{\text{SAS}}) + \ell_{\text{SA}}(\theta_{\text{SA}})$. At each iteration, we collect transitions from the source and (less frequently) target domain, storing the transitions in separate replay buffers. We then sample a batch of experience from both buffers to update the classifiers. We use the classifiers to modify the rewards from the *source* domain, and apply MaxEnt RL to this experience. We use SAC (Haarnoja et al., 2018) as our MaxEnt RL algorithm, but emphasize that DARC is applicable to any MaxEnt RL algorithm (e.g., on-policy, off-policy, and model-based). When training the classifiers, we add Gaussian input noise to prevent overfitting to the small number of target-domain transitions (see Fig. 7 for an ablation).

## 6   EXPERIMENTS

We start with a didactic experiment to build intuition for the mechanics of our method, and then evaluate on more complex tasks. Our experiments will show that DARC outperforms alternative approaches, such as directly applying RL to the target domain or learning importance weights. We will also show that our method can account for domain shift in the termination condition, and confirm the importance of learning two classifiers.

**Illustrative example.**   We start with a simple gridworld example, shown on the right, where we can apply our method without function approximation. The goal is to navigate from the top left to the bottom left. The real environment contains an obstacle (shown in red), which is not present in the source domain. If we simply apply RL on the source domain, we obtain

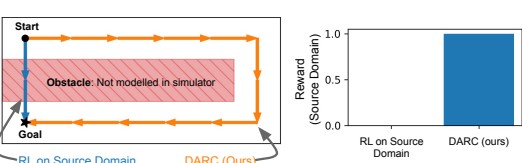

Figure 3: **Tabular example of off-dynamics RL**

a policy that navigates directly to the goal (blue arrows), and will fail when used in the target domain. We then apply our method: we collect trajectories from the source domain and real world to fit the two tabular classifiers. These classifiers give us a modified reward, which we use to learn a policy in the source domain. The modified reward causes our learned policy to navigate around the obstacle, which succeeds in the target environment.

**Visualizing the reward modification in stochastic domains.**   In our next experiment, we use an "archery" task to visualize how the modified reward accounts for differences in dynamics. The task, shown in Fig. 4, requires choosing an angle at which to shoot an arrow. The practice range (i.e., the source domain) is outdoors, with wind that usually blows from left to right. The competition range (i.e., the target domain) is indoors with no wind. The reward is the negative distance to the target. We

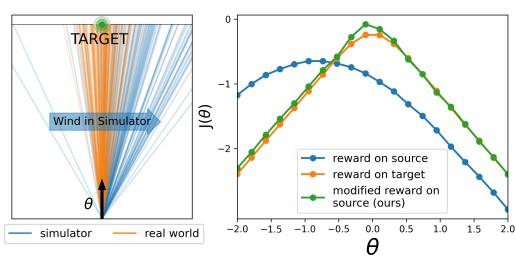

Figure 4: **Visualizing the modified reward**

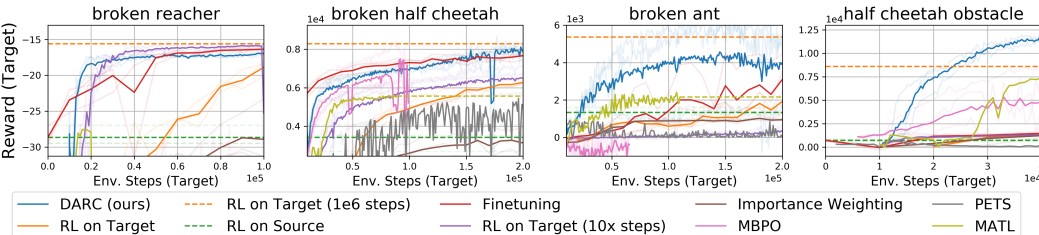

Figure 6: **DARC compensates for crippled robots and obstacles**: We apply DARC to four continuous control tasks: three tasks (broken reacher, half cheetah, and ant) which are crippled in the target domain but not the source domain, and one task (half cheetah obstacle) where the source domain omits the obstacle from the target domain. Note that naïvely ignoring the shift in dynamics (green dashed line) performs quite poorly, while directly learning on the crippled robot requires an order of magnitude more experience than our method.

plot the reward as a function of the angle in both domains in Fig. 4. The optimal strategy for the outdoor range is to compensate for the wind by shooting slightly to the left ($\theta = -0.8$), while the optimal strategy for the indoor range is to shoot straight ahead ($\theta = 0$). We estimate the modified reward function with DARC, and plot the modified reward in the windy outdoor range and indoor range. We aggregate across episodes using $J(\theta) = \log \mathbb{E}_{p(s'|\theta)}[\exp(r(s'))]$; see Appendix E.4 for details. We observe that maximizing the modified reward in the windy range does not yield high reward in the windy range, but does yield a policy that performs well in the indoor range.

**Scaling to more complex tasks.** We now apply DARC to the more complex tasks shown in Fig. 5. We define three tasks by crippling one of the joints of each robot in the target domain, but using the fully-functional robot in the source domain. We use three simulated robots taken from OpenAI Gym (Brockman et al., 2016): 7

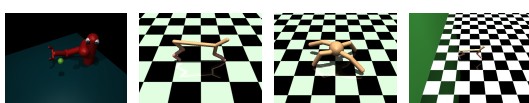

Figure 5: **Environments**: broken reacher, broken half cheetah, broken ant, and half cheetah obstacle.

DOF reacher, half cheetah, and ant. The broken reacher is based on the task described by Vemula et al. (2020). We also include a task where the shift in dynamics is external to the robot, by modifying the cheetah task to reward the agent for running both forward and backwards. It is easier to learn to run backwards, an obstacle in the target domain prevents the agent from running backwards. This "half cheetah obstacle" task does not entirely satisfy the assumption in Eq. 1 because transitions such as bouncing off the obstacle are only possible in the target domain, not the source domain. Nonetheless, the success of our method on this task illustrates that DARC can excel even in settings that do not satisfy the assumption in Eq. 1.

We compare our method to eight baselines. **RL on Source** and **RL on Target** directly perform RL on the source and target domains, respectively. The **Finetuning** baseline takes the result of running RL on the source domain, and further finetunes the agent on the target domain. The **Importance Weighting** baseline performs RL on importance-weighted samples from the source domain; the importance weights are $\exp(\Delta r)$. Recall that DARC collects many ($r = 10$) transitions in the source domain and performs many gradient updates for each single transition collected in the target domain (Alg. 1 Line 5). We therefore compared against a **RL on Target (10x)** baseline that likewise performs many ($r = 10$) gradient updates per transition in the target domain. Next, we compared against two recent model-based RL methods: **MBPO** (Janner et al., 2019) and **PETS** (Chua et al., 2018). Finally, we also compared against **MATL** (Wulfmeier et al., 2017b), which is similar in spirit to our method but uses a different modified reward.

We show the results of this experiment in Fig. 6, plotting the reward on the *target* domain as a function of the number of transitions in the *target* domain. In this figure, the transparent lines correspond to different random seeds, and the darker lines are the average of these random seeds. On all tasks, the RL on source baseline (shown as a dashed line because it observes no target transitions) performs considerably worse than the optimal policy from RL on the target domain, suggesting that good policies for the source domain are suboptimal for the target domain. Nonetheless, on three of the four tasks our method matches (or even surpasses) the asymptotic performance of doing RL on the target domain, despite never doing RL on experience from the target domain, and despite observing $5 - 10\times$ less experience from the target domain. On the broken reacher and broken half cheetah tasks, finetuning on the target domain performs on par with our method. On the simpler

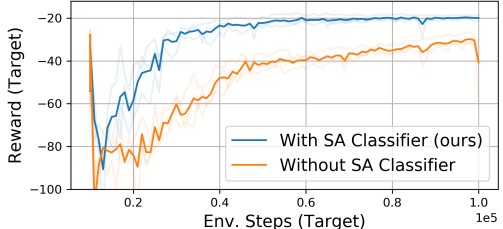 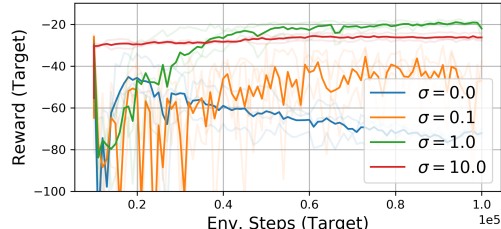

Figure 7: **Ablation experiments** *(Left)* DARC performs worse when only one classifier is used. *(Right)* Using input noise to regularize the classifiers boosts performance. Both plots show results for broken reacher; see Appendix D for results on all environments.

broken reacher task, just doing RL on the target domain with a large number of gradient steps works quite well (we did not tune this parameter for our method). While the model-based baselines (PETS and MBPO) also performed well on for low-dimensional tasks (broken reacher, broken half cheetah), they perform quite poorly on more challenging tasks like broken ant, supporting our intuition that classification is easier than learning a dynamics model in high dimensional tasks. Finally, DARC outperforms MATL on all tasks.

**Ablation Experiments.**   Our next experiment examines the importance of using two classifiers to estimate $\Delta r$. We compared our method to an ablation that does not learn the SA classifier, effectively ignoring the blue terms in Eq. 3. As shown in Fig. 7 (left), this ablation performs considerably worse than our method. Intuitively, this makes sense: we might predict that a transition came from the source domain not because the next state had higher likelihood under the source dynamics, but rather because the state or action was visited more frequently in the source domain. The second classifier used in our method corrects for this distribution shift.

Next, we examine the importance of input noise regularization in classifiers. As we observe only a handful of transitions from the target domain, we hypothesized that regularization would be important to prevent overfitting. We test this hypothesis in Fig. 7 (right) by training our method on the broken reacher environment with varying amounts of input noise. With no noise or little noise our method performs poorly (likely due to overfitting); too much noise also performs poorly (likely due to underfitting). We used a value of 1 in all our experiments, and did not tune this value. See Appendix D for more plots of both ablation experiments.

To gain more intuition for our method, we recorded the reward correction $\Delta r$ throughout training on the broken reacher environment. In this experiment, we ran RL on the source domain for 100k steps before switching to our method. Said another way, we ignored $\Delta r$ for the first 100k steps of training. As shown in Fig. 8, $\Delta r$ steadily decreases during these first 100k steps, suggesting that the agent is learning a strategy that takes transitions where the source domain and target domain have different dynamics: the agent is making use of its broken joint. After 100k steps, when we maximize the combination of task reward and $\Delta r$, we observe that $\Delta r$ increases, so the agent's transitions in the source domain are increasingly consistent with target domain dynamics. After around 1e6

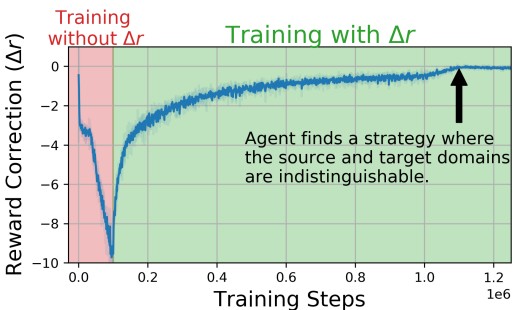

Figure 8: Without the reward correction, the agent takes transitions where the source domain and target domains are dissimilar; after adding the reward correction, the agent's transitions in the source domain are increasingly likely under the target domain.

training steps $\Delta r$ is zero: the agent has learned a strategy that uses transitions that are indistinguishable between the source and target domains.

**Safety emerges from domain adaptation to the termination condition.**   In many safety-critical applications, the real-world and simulator have different safeguards, which kick in to stop the agent and terminate the episode. For an agent to effectively transfer from the simulator to the real world, it cannot rely on safeguards which are present in one domain but not the other. Since this termination condition is part of the dynamics (White, 2017), we can readily apply DARC to this setting.

We use the humanoid shown in Fig. 9 for this experiment and set the task reward to 0. In the source domain episodes have a fixed length of 300 steps; in the target domain the episode terminates when the robot falls. The scenario mimics the real-world setting where robots have freedom to practice in a safe, cushioned, practice facility, but are preemptively stopped when they try to take unsafe actions in the real world. Our aim is for the agent to learn to avoid unsafe transitions in the source domain

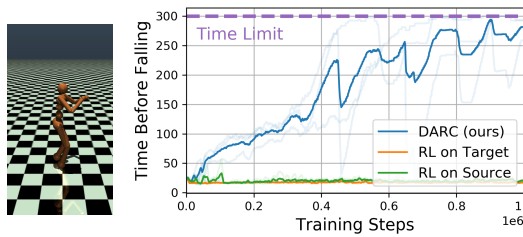

Figure 9: Our method accounts for domain shift in the termination condition, causing the agent to avoid transitions that cause termination in the target domain.

that would result in episode termination in the target domain. As shown in Fig. 9, our method learns to remain standing for nearly the entire episode. As expected, baselines that maximize the zero reward on the source and target domains fall immediately. While DARC was not designed as a method for safe RL (Tamar et al., 2013; Achiam et al., 2017; Eysenbach et al., 2017; Berkenkamp et al., 2017), this experiment suggests that safety may emerge automatically from DARC, without any manual reward function design.

**Comparison with Prior Transfer Learning Methods.** We are not the first work that modifies the reward function to perform transfer in RL (Koos et al., 2012), nor the first work to *learn* how to modify the reward function (Wulfmeier et al., 2017a). However, these prior works lack theoretical justification. In contrast, our approach maximizes a well-defined variational objective and our analysis guarantees that agents learned with our method will achieve similar rewards in the source and target domains. Our formal guarantees (Sec. 4)

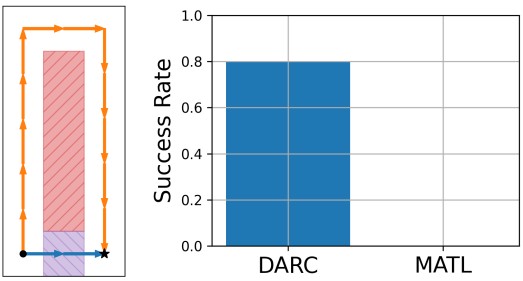

Figure 10: Comparison with MATL

do not apply to MATL (Wulfmeier et al., 2017b) because their classifier is not conditioned on the action. Indeed, our results on the four tasks in Fig. 6 indicate that DARC ourperforms MATL on all tasks. To highlight this difference, we compared DARC and MATL on a gridworld (right), where the source and target domains differed by assigning opposite effects to the "up" and "down" in the purple state in the source and target domains. We collected data from a uniform random policy, so the *marginal* distribution $p(s_{t+1} \mid s_t)$ was the same in the source and target domains, even though the dynamics $p(s_{t+1} \mid s_t, a_t)$ where different. In this domain, MATL fails to recognize that the source and target domains are different. DARC succeeds in this task for 80% of trials while MATL succeeds for 0% of trials. We conclude that the conditioning on the action, as suggested by our analysis, is especially important when using experience collected from stochastic policies.

## 7 DISCUSSION

In this paper, we proposed a simple, practical, and intuitive approach for domain adaptation to changing dynamics in RL. We motivate this method from a novel variational perspective on domain adaptation in RL, which suggests that we can compensate for differences in dynamics via the reward function. Moreover, we formally prove that, subject to a lightweight assumption, our method is guaranteed to yield a near-optimal policy for the target domain. Experiments on a range of control tasks show that our method can leverage the source domain to learn policies that will work well in the target domain, despite observing only a handful of transitions from the target domain.

**Limitations** The main limitation of our method is that the source dynamics must be sufficiently stochastic, an assumption that can usually be satisfied by adding noise to the dynamics, or ensembling a collection of sources. Empirically, we found that our method worked best on tasks that could be completed in many ways in the source domain, but some of these strategies were not compatible with the target dynamics. The main takeaway of this work is that inaccuracies in dynamics can be compensated for via the reward function. In future work we aim to use the variation perspective on domain adaptation (Sec. 4) to learn the dynamics for the source domain.

**Acknowledgements.** We thank Anirudh Vemula for early discussions; we thank Karol Hausman, Vincent Vanhoucke and anonymous reviews for feedback on drafts of this work. We thank Barry Moore for providing containers with MuJoCo and Dr. Paul Munro granting access to compute at CRC. This work is supported by the Fannie and John Hertz Foundation, University of Pittsburgh Center for Research Computing (CRC), NSF (DGE1745016, IIS1763562), ONR (N000141812861), and US Army. Any opinions, findings and conclusions or recommendations expressed in this material are those of the author(s) and do not necessarily reflect the views of the National Science Foundation.

**Contributions.** BE proposed the idea of using rewards to correct for dynamics, designed and ran many of the experiments in the paper, and wrote much of the paper. SA did the initial literature review, wrote and designed some of the DARC experiments and environments, developed visualizations of the modified reward function, and ran the MBPO experiments. SC designed some of the initial environments, helped with the implementation of DARC, and ran the PETS experiments. RS and SL provided guidance throughout the project, and contributed to the structure and writing of the paper.

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

# A  ADDITIONAL INTERPRETATIONS OF THE REWARD CORRECTION

This section presents four additional interpretations of the reward correction, $\Delta r$.

## A.1  CODING THEORY

The reward correction $\Delta r$ can also be understood from the perspective of coding theory. Suppose that we use a data-efficient replay buffer that exploits that fact that the next state $s_{t+1}$ is highly redundant with the current state and action, $s_t, a_t$. If we assume that the replay buffer compression has been optimized to store transitions from the target environment, (negative) $\Delta r$ is the number of additional bits (per transition) needed for our source replay buffer, as compared with our target replay buffer. Thus, an agent which maximizes $\Delta r$ will seek those transitions that can be encoded most efficiently, minimizing the size of the source replay buffer.

## A.2  MUTUAL INFORMATION

We can gain more intuition in the modified reward by writing the expected value of $\Delta r$ from Eq. 3 in terms of mutual information:

$$\mathbb{E}[\Delta r(s_t, a_t, s_{t+1})] = I(s_{t+1}; \text{target} \mid s_t, a_t) - I(s_{t+1}; \text{source} \mid s_t, a_t).$$

The mutual information $I(s_{t+1}; \text{target} \mid s_t, a_t)$ reflects how much better you can predict the next state if you know that you are interacting with the target domain, instead of the source domain. Our approach does exactly this, rewarding the agent for taking transitions that provide information about the target domain while penalizing transitions that hint to the agent that it is interacting with a source domain rather than the target domain: we don't want our are agent to find bugs in the Matrix.

## A.3  LOWER BOUND ON THE RISK-SENSITIVE REWARD OBJECTIVE.

While we derived DARC by minimizing a reverse KL divergence (Eq. 2), we can also show that DARC maximizes a lower bound on a risk-sensitive reward objective (Mihatsch & Neuneier, 2002):

$$
\log \mathbb{E}_{\substack{s' \sim p_{\text{target}}(s'|s,a), \\ a \sim \pi(a|s)}} \left[ \exp \left( \sum_t r(s_t, a_t) \right) \right]
$$

$$
= \log \mathbb{E}_{\substack{s' \sim p_{\text{source}}(s'|s,a), \\ a \sim \pi(a|s)}} \left[ \left( \prod_t \frac{p_{\text{target}}(s_{t+1} \mid s_t, a_t)}{p_{\text{source}}(s_{t+1} \mid s_t, a_t)} \right) \exp \left( \sum_t r(s_t, a_t) \right) \right]
$$

$$
= \log \mathbb{E}_{\substack{s' \sim p_{\text{source}}(s'|s,a), \\ a \sim \pi(a|s)}} \left[ \exp \left( \sum_t r(s_t, a_t) + \underbrace{\log p_{\text{target}}(s_{t+1} \mid s_t, a_t) - \log p_{\text{source}}(s_{t+1} \mid s_t, a_t)}_{\Delta r(s_t, a_t, s_{t+1})} \right) \right]
$$

$$\tag{4}$$

$$
\geq \mathbb{E}_{\substack{s' \sim p_{\text{source}}(s'|s,a), \\ a \sim \pi(a|s)}} \left[ \sum_t r(s_t, a_t) + \Delta r(s_t, a_t, s_{t+1}) \right]. \tag{5}
$$

The inequality on the last line is an application of Jensen's inequality. One interesting question is when it would be preferable to maximize Eq. 4 rather than Eq. 5. While Eq. 5 provides a loser bound on the risk sensitive objective, empirically it may avoid the risk-seeking behavior that can be induced by risk-sensitive objectives. We leave the investigation of this trade-off as future work.

## A.4  A CONSTRAINT ON DYNAMICS DISCREPANCY

Our method regularizes the policy to visit states where the transition dynamics are similar between the source domain and target domain:

$$
\max_\pi \mathbb{E}_{\substack{a \sim \pi(a|s) \\ s' \sim p(s'|s,a)}} \left[ \sum_t r(s_t, a_t) + \underbrace{\log p_{\text{target}}(s_{t+1} \mid s_t, a_t) - \log p_{\text{source}}(s_{t+1} \mid s_t, a_t)}_{-D_{\text{KL}}(p_{\text{source}} \parallel p_{\text{target}})} + \mathcal{H}_\pi[a_t \mid s_t] \right].
$$

This objective can equivalently be expressed as applying MaxEnt RL to only those policies which avoid exploiting the dynamics discrepancy. More precisely, the KKT conditions guarantee that there exists a positive constant $\epsilon > 0$ such that our objective is equivalent to the following constrained objective:

$$\max_{\pi \in \Pi_{\text{DARC}}} \mathbb{E}_{\substack{a \sim \pi(a|s) \\ s' \sim p(s'|s,a)}} \left[ \sum_t r(s_t, a_t) + \mathcal{H}_\pi[a_t \mid s_t] \right],$$

where $\Pi_{\text{DARC}}$ denotes the set of policies that do not exploit the dynamics discrepancy:

$$\Pi_{\text{DARC}} \triangleq \left\{ \pi \middle| \mathbb{E}_{\substack{a \sim \pi(a|s) \\ s' \sim p(s'|s,a)}} \left[ \sum_t D_{\text{KL}}(p_{\text{source}}(s_{t+1} \mid s_t, a_t) \parallel p_{\text{target}}(s_{t+1} \mid s_t, a_t)) \right] \leq \epsilon \right\}. \quad (6)$$

One potential benefit of considering our method as the unconstrained objective is that it provides a principled method for increasing or decreasing the weight on the $\Delta r$ term, depending on how much the policy is currently exploiting the dynamics discrepancy. We leave this investigation as future work.

# B  PROOFS OF THEORETICAL GUARANTEES

In this section we present our analysis showing that maximizing the modified reward $r + \Delta r$ in the source domain yields a near-optimal policy for the target domain, subject to Assumption 1. To start, we show that maximizing the modified reward in the source domain is equivalent to maximizing the unmodified reward, subject to the constraint that the policy not exploit the dynamics:

**Lemma B.1.** *Let a reward function $r(s, a)$, source dynamics $p_{source}(s' \mid s, a)$, and target dynamics $p_{target}(s' \mid s, a)$ be given. Then there exists $\epsilon > 0$ such the optimization problem in Eq. 2 is equivalent to*

$$\max_{\pi \in \Pi_{\text{no exploit}}} \mathbb{E}_{p_{source}, \pi} \left[ \sum r(s_t, a_t) + \mathcal{H}_\pi[a_t \mid s_t] \right],$$

*where $\Pi_{no\ exploit}$ denotes the set of policies that do not exploit the dynamics:*

$$\Pi_{no\ exploit} \triangleq \left\{ \mathbb{E}_{\substack{a \sim \pi(a|s) \\ s' \sim p(s'|s,a)}} \left[ \sum_t D_{\text{KL}}(p_{source}(s_{t+1} \mid s_t, a_t) \parallel p_{target}(s_{t+1} \mid s_t, a_t)) \right] \leq \epsilon \right\}.$$

The proof is a straightforward application of the KKT conditions. This lemma says that maximizing the modified reward can be equivalently viewed as restricting the set of policies to those that do not exploit the dynamics. Next, we will show that policies that do not exploit the dynamics have an expected (entropy-regularized) reward that is similar in the source and target domains:

**Lemma B.2.** *Let policy $\pi \in \Pi_{no\ exploit}$ be given, and let $R_{max}$ be the maximum (entropy-regularized) return of any trajectory. Then the following inequality holds:*

$$\left| \mathbb{E}_{p_{source}} \left[ \sum r(s_t, a_t) + \mathcal{H}_\pi[a_t \mid s_t] \right] - \mathbb{E}_{p_{target}} \left[ \sum r(s_t, a_t) + \mathcal{H}_\pi[a_t \mid s_t] \right] \right| \leq 2R_{max}\sqrt{\epsilon/2}.$$

This Lemma proves that all policies in $\Pi_{\text{no exploit}}$ satisfy the same condition as the optimal policy (Assumption 1).

*Proof.* To simplify notation, define $\tilde{r}(s, a) = r(s, a) - \log \pi(a \mid s)$. We then apply Holder's inequality and Pinsker's inequality to obtain the desired result:

$$\mathbb{E}_{p_{\text{source}}} \left[ \sum \tilde{r}(s_t, a_t) \right] - \mathbb{E}_{p_{\text{target}}} \left[ \sum \tilde{r}(s_t, a_t) \right] = \sum_\tau (p_{\text{source}}(\tau) - p_{\text{target}}(\tau)) \left( \sum \tilde{r}(s_t, a_t) \right)$$

$$\leq \| \sum \tilde{r}(s_t, a_t) \|_\infty \cdot \| p_{\text{source}}(\tau) - p_{\text{target}}(\tau) \|_1$$

$$\leq \left( \max_\tau \sum r(s_t, a_t) \right) \cdot 2\sqrt{\frac{1}{2} D_{\text{KL}}(p_{\text{source}}(\tau) \parallel p_{\text{target}}(\tau))}$$

$$\leq 2R_{\text{max}}\sqrt{\epsilon/2}.$$

$\square$

We restate our main result:

**Theorem 4.1** (Repeated from main text). *Let $\pi_{DARC}^*$ be the policy that maximizes the modified (entropy-regularized) reward in the source domain, let $\pi^*$ be the policy that maximizes the (unmodified, entropy-regularized) reward in the target domain, and assume that $\pi^*$ satisfies Assumption 1. Then $\pi_{DARC}^*$ receives near-optimal (entropy-regularized) reward on the target domain:*

$$\mathbb{E}_{p_{target},\pi_{DARC}^*}\left[\sum r(s_t,a_t) + \mathcal{H}[a_t \mid s_t]\right] \geq \mathbb{E}_{p_{target},\pi^*}\left[\sum r(s_t,a_t) + \mathcal{H}[a_t \mid s_t]\right] - 4R_{max}\sqrt{\epsilon/2}.$$

*Proof.* Assumption 1 guarantees that the optimal policy for the target domain, $\pi^*$, lies within $\Pi_{\text{no exploit}}$. Among all policies in $\Pi_{\text{no exploit}}$, $\pi_{\text{DARC}}^*$ is (by definition) the one that receives highest reward on the source dynamics, so

$$\mathbb{E}_{p_{\text{source}},\pi_{\text{DARC}}^*}\left[\sum r(s_t,a_t) + \mathcal{H}[a_t \mid s_t]\right] \geq \mathbb{E}_{p_{\text{source}},\pi^*}\left[\sum r(s_t,a_t) + \mathcal{H}[a_t \mid s_t]\right].$$

Since the both $\pi_{\text{DARC}}^*$ and $\pi^*$ lie inside the constraint set, Lemma B.2 dictates that their rewards on the target domain differ by at most $2R_{\max}\sqrt{\epsilon/2}$ from their source domain rewards. In the worst case, the reward for $\pi_{\text{DARC}}^*$ decreases by this amount and the reward for $\pi^*$ increases by this amount:

$$\mathbb{E}_{p_{\text{source}},\pi_{\text{DARC}}^*}\left[\sum r(s_t,a_t) + \mathcal{H}[a_t \mid s_t]\right] \leq \mathbb{E}_{p_{\text{target}},\pi_{\text{DARC}}^*}\left[\sum r(s_t,a_t) + \mathcal{H}[a_t \mid s_t]\right] + 2R_{\max}\sqrt{\epsilon/2}$$

$$\mathbb{E}_{p_{\text{source}},\pi^*}\left[\sum r(s_t,a_t) + \mathcal{H}[a_t \mid s_t]\right] \geq \mathbb{E}_{p_{\text{target}},\pi^*}\left[\sum r(s_t,a_t) + \mathcal{H}[a_t \mid s_t]\right] - 2R_{\max}\sqrt{\epsilon/2}$$

Substituting these inequalities on the LHS and RHS of Eq. B and rearranging terms, we obtain the desired result. □

## C   THE SPECIAL CASE OF AN OBSERVATION MODEL

To highlight the relationship between domain adaptation of dynamics versus observations, we now consider a special case. In this subsection, we will assume that the state $s_t \triangleq (z_t, o_t)$ is a combination of the system latent state $z_t$ (e.g., the poses of all objects in a scene) and an observation $o_t$ (e.g., a camera observation). We will define $q(o_t \mid z_t)$ and $p(o_t \mid z_t)$ as the *observation models* for the source and target domains. In this special case, we can decompose the KL objective (Eq. 2) into three terms:

$$D_{\text{KL}}(q \parallel p) = -\mathbb{E}_q\left[\sum_t \underbrace{r(s_t,a_t) + \mathcal{H}_\pi[a_t \mid s_t]}_{\text{MaxEnt RL objective}} + \underbrace{\log p_{\text{target}}(o_t \mid z_t) - \log p_{\text{source}}(o_t \mid z_t)}_{\text{Observation Adaptation}} \right.$$
$$\left. + \underbrace{\log p_{\text{target}}(z_{t+1} \mid z_t, a_t) - \log p_{\text{source}}(z_{t+1} \mid z_t, a_t)}_{\text{Dynamics Adaptation}} \right].$$

Prior methods that perform observation adaptation (Bousmalis et al., 2018; Gamrian & Goldberg, 2018) effectively minimize the observation adaptation term,[1] but ignore the effect of dynamics. In contrast, the $\Delta r$ reward correction in our method provides one method to address both dynamics and observations. These approaches could be combined; we leave this as future work.

---

[1]Tiao et al. (2018) show that observation adaptation using CycleGan (Zhu et al., 2017a) minimizes a Jensen-Shannon divergence. Assuming sufficiently expressive models, the Jensen-Shannon divergence and the reverse KL divergence above have the same optimum.

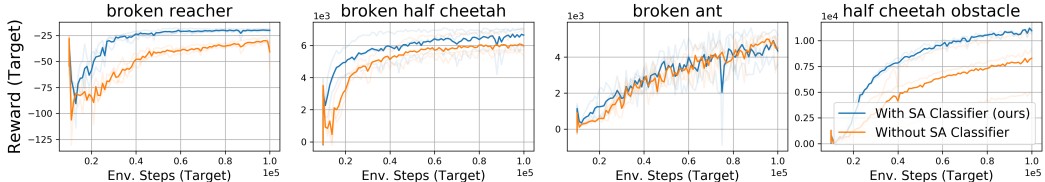

Figure 11: **Importance of using two classifiers**: Results of the ablation experiment from Fig. 7 (left) on all environments.

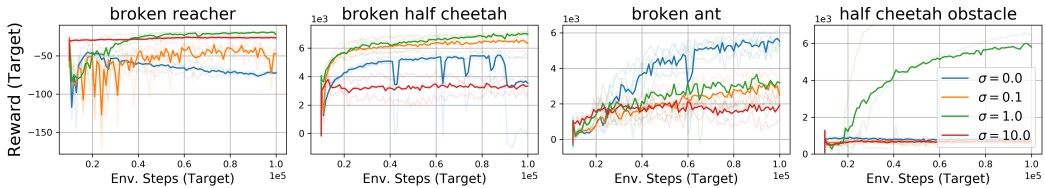

Figure 12: **Importance of regularizing the classifiers**: Results of the ablation experiment from Fig. 7 (right) on all environments.

## D    ADDITIONAL EXPERIMENTS

Figures 11 and 12 show the results of the ablation experiment from Fig. 7 run on all environments. The results support our conclusion in the main text regarding the importance of using two classifiers and using input noise. Figure 13 is a copy of Fig. 8 from the main text, modified to also show the agent's reward on the target domain. We observe that the reward does not start increasing until we start using DARC.

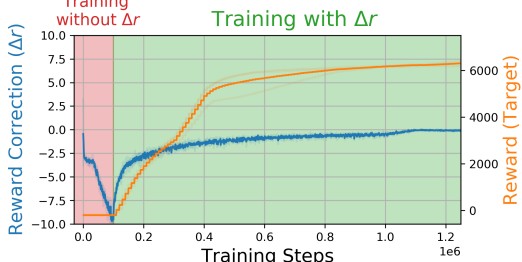

Figure 13: Copy of Fig. 8 overlaid with the target domain reward.

## E    EXPERIMENT DETAILS AND HYPERPARAMETERS

Our implementation of DARC is built on top of the implementation of SAC from Guadarrama et al. (2018). Unless otherwise specified, all hyperparameters are taken from Guadarrama et al. (2018). All neural networks (actor, critics, and classifiers) have two hidden layers with 256-units each and ReLU activations. Since we ultimately will use the *difference* in the predictions of the two classifiers, we use a residual parametrization for the SAS classifier $q(\text{target} \mid s_t, a_t, s_{t+1})$. Using $f_{\text{SAS}}(s_t, a_t, s_{t+1}), f_{\text{SA}}(s_t, a_t) \in \mathbb{R}^2$ to denote the outputs of the two classifier networks, we compute the classifier predictions as follows:

$$q_{\theta_{\text{SA}}}(\cdot \mid s_t, a_t) = \text{SOFTMAX}(f_{\text{SA}}(s_t, a_t))$$
$$q_{\theta_{\text{SAS}}}(\cdot \mid s_t, a_t, s_{t+1}) = \text{SOFTMAX}(f_{\text{SAS}}(s_t, a_t, s_{t+1}) + f_{\text{SA}}(s_t, a_t))$$

For the SAS classifier we propagate gradients back through both networks parameters, $\theta_{\text{SAS}}$ and $\theta_{\text{SA}}$. Both classifiers use Gaussian input noise with $\sigma = 1$. Optimization of all networks is done with Adam (Kingma & Ba, 2014) with a learning rate of 3e-4 and batch size of 128. Most experiments with DARC collected 1 step in the target domain every 10 steps in the source domain (i.e., $r = 10$). The one exception is the half cheetah obstacle domain, where we tried increasing $r$ beyond 10 to 30, 100, 300, and 1000. We found a large benefit from increasing $r$ to 30 and 100, but did not run the other experiments long enough to draw any conclusions. Fig. 6 uses $r = 30$ for half cheetah obstacle. We did not tune this parameter, and expect that tuning it would result in significant improvements in sample efficiency.

We found that DARC was slightly more stable if we warm-started the method by applying RL on the source task *without* $\Delta r$ for the first $t_{\text{warmup}}$ iterations. We used $t_{\text{warmup}} = 1e5$ for all tasks except the broken reacher, where we used $t_{\text{warmup}} = 2e5$. This discrepancy was caused by a typo in an experiment, and subsequent experiments found that DARC is relatively robust to different values of $t_{\text{warmup}}$; we did not tune this parameter.

## E.1 BASELINES

The **RL on Source** and **RL on Target** baselines are implemented identically to our method, with the exception that $\Delta r$ is not added to the reward function. The **RL on Target (10x)** is identical to RL on Target, with the exception that we take 10 gradient steps per environment interaction (instead of 1). The **Importance Weighting** baseline estimates the importance weights as $p_{\text{target}}(s_{t+1} \mid s_t, a_t)/p_{\text{source}}(s_{t+1} \mid s_t, a_t) \approx \exp(\Delta r)$. The importance weight is used to weight transitions in the SAC actor and critic losses.

**PETS (Chua et al., 2018)**  The **PETS** baseline is implemented using the default configurations used by (Chua et al., 2018) for the environments evaluated. The `broken-half-cheetah` environment uses the hyperparameters as used by the `half-cheetah` environment in (Chua et al., 2018). The `broken-ant` environment uses the same set of hyperparameters, namely: task horizon = 1000, number of training iterations = 300, number of planning (real) steps per iteration = 30, number of particles to be used in particle propagation methods = 20. The PETS codebase can be found at `https://github.com/kchua/handful-of-trials`.

**MBPO (Janner et al., 2019)**  We used the authors implementation with the default hyperparameters: `https://github.com/JannerM/mbpo`. We kept the environment configurations the same as their default unmodified MuJoCo environments, except for the domain and task name. We added our custom environment xmls in `mbpo/env/assets/` folder, and their corresponding environment python files in the `mbpo/env/` folder. Their static files were added under `mbpo/static/`. These environments can be registered as gym environments in the init file under `mbpo_odrl/mbpo/env/` or can be initialized directly in `softlearning/environments/adapters/gym adapter.py`. We set the time limit to `max_episode_steps=1000` for the Broken Half Cheetah, Broken Ant and Half Cheetah Obstacle environments and to 100 for the Broken Reacher environment.

## E.2 ENVIRONMENTS

**Broken Reacher**  This environment uses the 7DOF robot arm from the Pusher environment in OpenAI Gym. The observation space is the position and velocities of all joints and the goal. The reward function is

$$ r(s, a) = -\frac{1}{2}\|s_{\text{end effector}} - s_{\text{goal}}\|_2 - \frac{1}{10}\|a\|_2^2, $$

and episodes are 100 steps long. In the target domain the 2nd joint (0-indexed) is broken: zero torque is applied to this joint, regardless of the commanded torque.

**Broken Half Cheetah**  This environment is based on the HalfCheetah environment in OpenAI Gym. We modified the observation to include the agent's global X coordinate so the agent can infer its relative position to the obstacle. Episodes are 1000 steps long. In the target domain the 0th joint (0-indexed) is broken: zero torque is applied to this joint, regardless of the commanded torque.

**Broken Ant**  This environment is based on the Ant environment in OpenAI Gym. We use the standard termination condition and cap the maximum episode length at 1000 steps. In the target domain the 3rd joint (0-indexed) is broken: zero torque is applied to this joint, regardless of the commanded torque.

In all the broken joint environments, we choose which joint to break to computing which joint caused the "RL on Source" baseline to perform worst on the target domain, as compared with the "RL on Target" baseline.

**Half Cheetah Obstacle**    This environment is based on the HalfCheetah environment in OpenAI Gym. Episodes are 1000 steps long. We modified the standard reward function to use the absolute value in place of the velocity, resulting the following reward function:

$$r(s, a) = s_{\text{x vel}} \cdot \Delta t - \|a\|_2^2,$$

where $s_{\text{x vel}}$ is the velocity of the agent along the forward-aft axis and $\Delta t = 0.01$ is the time step of the simulator. In the target domain, we added a wall at $x = -3m$, roughly 3 meters behind the agent.

**Humanoid**    Used for the experiment in Fig. 9, we used a modified version of Humanoid from OpenAI Gym. The source domain modified this environment to ignore the default termination condition and instead terminate after exactly 300 time steps. The target domain uses the unmodified environment, which terminates when the agent falls.

### E.3    FIGURES

Unless otherwise noted, all experiments were run with three random seeds. Figures showing learning curves (Figures 6, 7, 8, 11, and 12) plot the *mean* over the three random seeds, and also plot the results for each individual random seed with semi-transparent lines.

### E.4    ARCHERY EXPERIMENT

We used a simple physics model for the archery experiment. The target was located 70m North of the agent, and wind was applied along the East-West axis. The system dynamics:

$$s_{t+1} = 70\sin(\theta) + f/\cos(\theta)^2 \qquad \begin{cases} f \sim \mathcal{N}(\mu = 1, \sigma = 1) & \text{in the source domain} \\ f \sim \mathcal{N}(\mu = 0, \sigma = 0.3) & \text{in the target domain} \end{cases}$$

We trained the classifier by sampling $\theta \sim \mathcal{U}[-2, 2]$ (measured in degrees) for 10k episodes in the source domain and 10k episodes in the target domain. The classifier was a neural network with 1 hidden layer with 32 hidden units and ReLU activation. We optimized the classifier using the Adam optimizer with a learning rate of 3e-3 and a batch size of 1024. We trained until the validation loss increased for 3 consecutive epochs, which took 16 epochs in our experiment. We generated Fig. 4 by sampling 10k episodes for each value of $\theta$ and aggregating the rewards using $J(\theta) = \log \mathbb{E}_{p(s'|\theta)}[\exp(r(s'))]$. We found that aggregating rewards by taking the mean did not yield meaningful results, perhaps because the mean corresponds to a (possibly loose) lower bound on $J$ (see Appendix A.3).

