# OpenReview forum: "Off-Dynamics Reinforcement Learning: Training for Transfer with Domain Classifiers"
_ICLR.cc/2021/Conference — ICLR 2021 Poster_

### Official Review · AnonReviewer2 · 2020-10-25
**review2**

**Rating:** 7
**Confidence:** 3

**Review:**

This paper presents a method to do domain adaptation. The idea is to modify the reward function in the source domain so the learned policy can be optimal in the target domain. This is accomplished by learning classifiers that distinguish transition in the source domain from transition in the target domain.

The idea is introduced intuitively and justified via formal analysis and it seems to effectively solve the tasks presented, but I have a few questions that need clarification before I make a final decision.

(1) The assumption that the target transition will have non-zero probability in the source transition seems to be violated in some of the examples. For example, if the action for the particular broken joint is non-zero, the transition in the target domain will be entirely different from the source domain since the simulator is deterministic. I don't see how this will satisfy the assumption.

(2)Further comment on this assumption. For the motivating example of driving on roads with different frictions. Again transition in the target domain (icy road) has almost zero probability in the source domain.

(3) About Assumption 1 in the paper. What is $\epsilon$? What is $R_{max}$? I see this is defined in the appendix, but please define it in the main text.  And this assumption seems to become lemmaB2 (or look very similar to it), this is quite confusing. And for LemmaB2, there seems to be a typo in the first inequality, I assume the second p_source should be p_target, otherwise I can't see the difference between the two expectation. And what is the $p$ and $q$ in the first line of the proof?

(4) Further question on the assumption, since R_max is the maximum total reward possible for a trajectory, this can be a large number (for example, for half cheetah it is something like 8000). I don't see why this assumption is meaningful.

(5) For the half cheetah obstacle examples, the problem is not a MDP anymore since the obstacles is placed at some specific position and the policy doesn't have access to the position information as far as I know.

(6) The experiment for humanoid standing, while interesting, seems unrealistic. In real world scenario, the robot should at least be rewarded an alive bonus for avoiding the termination condition.

---

> ### Author Response · Authors · 2020-11-20
> **Response to R2**
>
> We thank the reviewer for the valuable review. We'll answer the questions below:
> 1. Yes, that's a great point. While we needed the assumption in Eq 1 to provide _theoretical_ guarantees for our method, our experiments illustrate that DARC works well even on environments that don't meet this assumption. We have revised the paper to discuss this.
> 2. What's crucial for the success of DARC is there is some way of completing the task where the dynamics will look similar between the source and target domain. Driving on an icy road, while maintaining a low speed and avoiding rapid acceleration, it's actually quite hard to tell if the car is on ice or not. To demonstrate this, we ran additional experiments showing that DARC can successfully learn behaviors that transfer to icy environments (see general comment above and videos+plots on the website: https://darc-anonymous.github.io/).
> 3. We've added a definition of $\epsilon$ and $R_\text{max}$ to the main text and clarified that $p$ and $q$ refer to $p_\text{target}$ and $p_\text{source}$ respectively. The difference between Lemma B.2 and Assumption 1 is that Assumption 1 only applies to the optimal policy, where Lemma B.2 proves that the same inequality holds for all policies in the set $\Pi_\text{no exploit}$. We have added a sentence describing this after the statement of Lemma B.2 in the Appendix.
> 4. Both Assumption 1 and Theorem 4.1 can be understood as measuring the expected reward relative to the maximum possible reward. To see this, we can divide both sides of the inequalities in each statement by $R_\text{max}$.
> 5. For the half cheetah obstacle task, we modified the standard observation to also include the agent's global X position. We've added this detail to Appendix E.2.
> 6. Determining that "alive" bonus can be challenging in real-world domains, especially since safeguards that would trigger this "alive" bonus occur at many levels of the robotics stack, from power limiters to safety barricades. Manually enumerating all the safeguards and instrumenting each to provide a "not alive" penalty when triggered would require considerable engineering effort. In contrast, our method achieves a similar effect without requiring _any_ engineering effort.

---

> > ### Comment · AnonReviewer2 · 2020-11-21
> > **thanks for clarification**
> >
> > Thanks for the clarification, I updated the score accordingly.

---

### Official Review · AnonReviewer4 · 2020-10-27
**DARC is cool and the motivation is solid, but the assumption 1 is quite restrictive**

**Rating:** 6
**Confidence:** 4

**Review:**


## Summary

The paper introduces DARC, a domain transfer algorithm motivated by maximum entropy RL. By introducing classifiers for the target and source domain, the reward function in the source domain can be modified such that it restricts the behavior of the optimized policy to transitions that reflect the target domain. In this way, the method achieves good domain transfer without having an explicit model.

Additionally, the authors provide a lower bound on the performance of such a policy in the target domain.

## Overall Recommendation

Although I believe the method to be quite restricted by the assumptions that it makes, overall, I recommend acceptance, but it could be stronger after clearing up a few things.

## Writeup

I really like the writeup, the connection to the related literature is clear and it is clear what is the goal of the method. The method is explained well and I didn't have any problems following the text.

## Pros

The method is very intuitive and theoretically well justified. The authors provide experiments and compare against different model-free settings and a model-based baseline. I also like the connection to safety, namely that this method ensures safety by enforcing behavior that is far away from violating the safety constraints.

## Cons

The experiments are still a bit dissatisfying, because I would like to see an experiment where transfer happens to the real system. Also, I can see a benefit in improving the simulators or the source domain to reflect the target domain, I don't think that taking the source domain as-is is the way to go, because this is also a waste of data from the target domain.

I take note that data collection from the target domain is nevertheless necessary, in order to fit the classifiers. Might as well improve the simulator(model) in the source domain.

Assumption 1 is quite restrictive.

## Comments/Questions

The assumption on the reward maximizing policy from the target domain is quite strong, this also translates to an assumption about how much the dynamics are allowed to change. It would seem more satisfying to me to take into account the transition dynamics errors in deriving the bound, such as done in MBPO[1].

I believe that the way you chose to construct your experiments reflect your assumptions nicely. For the broken cheetah/ant/reacher, a policy that works with less degrees of freedom would more-or-less obviously work with more (satisfying assumption  1). I wonder how would your method perform with not so obvious changes in dynamics - changes in mass, friction etc, perhaps even adversarial changes in dynamics?

Normally, when doing sim2real, the dynamics change in a different way than what is shown in the experiments.

Throwing away data from the target domain and not improving the source domain seems like the wrong thing to do,  ideally you would reuse the data from the target domain to improve the source domain.

Furthermore, as far as I know,  MBPO  doesn't work on transfer, but actually learns the simulator for the environment (the model). Obviously if the data for fitting the model only comes from the source domain, the model would overfit to the source domain but  stick to the regions that it's certain about. The data from the target domain should be reused  for MBPO, or it should be given the same chance as RL trained on target.  Was this done in this way? The results for MBPO seem a bit too disappointing to be true.

A further separating thing is that, you assume that you basically have a simulator that "encompasses" the target domain, i.e. policies good in the target domain are still valid in the source domain, this is not needed by MBPO or the model-free variants, and it obviously helps the method a lot. In the broken environments case, if you would switch the target and source domains and then evaluate against the baselines, I believe that the baselines would be better or at least on-par.

-------

[1] Janner, Michael, et al. "When to trust your model: Model-based policy optimization." Advances in Neural Information Processing Systems. 2019.

---

> ### Author Response · Authors · 2020-11-20
> **Response to R4**
>
> Thank you for the review and suggestions for improvement. Our understanding is that the reviewer’s three main concerns are (1) Assumption 1; (2) comparing against baselines that update the simulator using experience from the target domain; and (3) additional experiments with less obvious changes to the dynamics, such as changing friction. Below, we argue why Assumption 1 is reasonable and argue that we have already compared against baselines that update the simulator. We have run additional experiments showing that DARC performs well on "icy" environments where friction changes (see general comment and website: https://darc-anonymous.github.io/). We ask the reviewer to revisit the review in light of these arguments.
>
> **Updating the simulator with target domain experience**: The model-based RL methods that we compare against in Fig 6 (MBPO and PETS) do precisely this. They learn a dynamics model using only experience gathered in the target domain. All methods have only limited access to experience from the target domain, so we attribute the poor performance of MBPO and PETS to the fact that they are trained using so little experience. We experimented with combining DARC with these model-based RL methods; preliminary results were poor, but we think this is an exciting direction for future work.
>
> **Assumption 1** says that policies that perform well in the target domain also perform well in the source domain. This assumption encodes the intuition that the optimal policy for the target domain won't exploit inaccuracies in the simulator. Note that the most similar prior work [Wulfmeier '17] does not discuss this assumption because this prior work does not aim to provide theoretical guarantees. In contrast, we show in Theorem 4.1 that our method learns a near-optimal policy for the target domain, despite practicing only in the source domain. Our _theoretical_ results relies on this assumption; without this assumption, no guarantee is possible.

---

### Official Review · AnonReviewer3 · 2020-10-27
**Well-written and well-motivated, but needs comparisons to stronger baselines, more evidence of robustness to dynamics, and discussion of limitations.**

**Rating:** 6
**Confidence:** 4

**Review:**

Summary: This paper introduces DARC, an RL approach that aims to transfer from a source environment to a target environment with different dynamics. DARC utilizes an adversarial-like reward that distinguishes between source and target domains. This reward is used to augment the environmental reward. DARC is evaluated against several baselines in mujoco tasks.


Overall, the paper is well-written and the motivation is clear. The approach seems reproducible, as pseudocode is provided along with training details and hyperparameters. The technique seems simple and easy to implement. The paper also includes some theoretical guarantees for the approach.


However, I believe this paper is borderline and I lean towards reject for a few reasons. The quick summary is that stronger baselines could have been included, it is unclear how well the approach can handle large (or even medium) differences in dynamics, and the approach does not seem robust to choices of source and target domains.


While the method was evaluated against several baselines, I believe there are stronger methods it could have been compared to. In particular, there are quite a few imitation learning approaches that aim to learn in environments with different dynamics than the demonstrator, which is similar to the idea of transfer [1-3]. Each of these approaches has been shown to work in mujoco environments with disabled joints and hence would be good baselines. While MATL is a reasonable comparison, it was not evaluated in such environments in the original paper.


Furthermore, using adversarial-based rewards to classify whether a trajectory is from the agent or source is not a new concept. It originated from GAIL [4] and has since seen several variants, including works that learn across different variants of the domain [5-7]. The paper should be more clear in expressing that domain classification in reinforcement learning is not a new concept.


The requirement that $P_{target}(s_{t+1}|s_t, a_t) > 0$ ⇒ $P_{source}(s_{t+1}|s_t, a_t) > 0$ is a weak point of the approach. It is unclear if the lower limit is 0 or some larger number in practice. This is important given the claim that DARC is for transfer in environments with different dynamics.  If the dynamics are very different, then I would expect the discriminative reward to do poorly because it would be easy to distinguish between source and target transitions. An interesting experiment would be to see how the discriminator behaves as the discrepancies in dynamics increases.


Finally, it seems that one needs to get a bit lucky in the source and target choice. The example shown in fig 3. shows that if there were an obstacle in the target domain and no obstacle in the source domain, then the agent would need to go around this obstacle. But if the obstacle were in the source domain, I would expect the approach to be suboptimal in the target domain since the reward function would encourage it to go around an invisible object.


Some concrete steps for improving the paper would be to include more discussion as well as comparisons to approaches that seem equipped to handling different dynamics, demonstrating the impact of differences in dynamics on the approach, and making the limitations of the approach more clear.


Other comments/thoughts:


The acronym DARC is introduced in the theorem. It may be better to introduce beforehand


What does the environment reward in Fig. 7 look like? It would be interesting to see a similar jump in return


The introduction mentions a 111-dimensional ant task. Was this actually used?


“Our method performs 10× more gradient updates per environment step in the source domain” -- I don’t think this is explained in the paper?


Why does DARC sometimes have higher asymptotic reward than training from scratch?

Typos


higher-dimensional tasks. broadly-applicable approach for learning from inaccurate models.


that most prior work -> than most prior work


the perform quite poorly -> they perform quite poorly


so the agent is penalized -> the agent is penalized


This result that


safegaurds -> safeguards


[1] Adversarial Imitation via Variational Inverse Reinforcement Learning. Qureshi et al.


[2] Learning Robust Rewards with Adversarial Inverse Reinforcement Learning. Fu et al.


[3] State Alignment-Based Reinforcement Learning. Liu et al.


[4] Generative Adversarial Imitation Learning. Ho et al.


[5] Third-Person Imitation Learning. Stadie et al.


[6] Generative Adversarial Imitation from Observation. Torabi et al.


[7] Cross-domain Imitation Learning. Kim et al.

---

> ### Author Response · Authors · 2020-11-20
> **Response to R3**
>
> We thank the reviewer for the detailed review. The main aim of this response is clarify the role of Assumption 1 and to convince the reviewer that we have already compared against strong baselines. As suggested in the reviewer, we ran an additional experiment comparing the performance of DARC as the discrepancy between the source and target domains increases. Please see the general comment and website (https://darc-anonymous.github.io/) for details about this experiment. We kindly ask the reviewer to revisit their review in light of these clarifications and revisions.
>
> **Baselines**: We appreciate the suggestion to compare against imitation learning methods. Imitation learning methods typically solve a different problem than our method: while imitation learning methods require access to expert demonstrations (in some domain), our method doesn't require demonstrations. While the referenced imitation learning papers do consider two domains, our setting is different. In these papers, an expert collects experience in one environment and the agent practices and is evaluated in a second environment. In contrast, our paper considers the setting where the agent practices in one environment and is evaluated in a second environment. We have revised the paper to discuss how domain classifiers have also been used in imitation learning. The only prior work we are aware of that considers the same setting as our work is [Wulfmeier '17], which we have already compared against in our experiments. We'd welcome suggestions of other prior work that considers the same setting.
>
> **Assumptions and Limitations**: Please refer to the General Comment for a discussion of our assumption about the source and target domains. We have revised the paper to include this discussion, and added a paragraph about limitations to the end of the main text.
>
> We think the example offered in the "one needs to get a bit lucky" comment violates Assumption 1. Assumption 1 says that policies that perform well in the target domain also perform well in the source domain. If the source domain had an obstacle that wasn't included in the target domain, that the optimal policy for the target domain would perform poorly on the source domain, violating Assumption 1. This assumption encodes the intuition that the optimal policy for the target domain won't exploit inaccuracies in the simulator.
>
> **Writing and Clarifications**: We appreciate all the writing comments and typos and have addressed all of them in the revised paper.
> * Per your request, we have plotted the environment reward of the agent from Fig. 7, and added this in the Appendix as Fig 13.
> * We used the 111-dimensional ant for the "broken ant" task shown in Fig. 5 (second from right) and Fig 6 (second from right). We have also used a modified version of this 111-dimensional ant for some of the new "icy" experiments.
> * "10× more gradient updates per environment step" -- DARC takes 1 gradient step per source domain step. Since DARC collects 10 source domain steps per every 1 target domain step, it effectively takes 10 gradient steps per every 1 target domain step. We have revised the experiment section to discuss this detail.
> * Why does DARC sometimes have higher asymptotic reward than training from "scratch?" -- This is in reference to DARC outperforming the "RL on Target (1e6 steps)" line on the half cheetah obstacle task, right? We hypothesize that DARC performs better on this task because the $\Delta r$ term provides useful reward shaping, thereby accelerating learning.

---

### Official Review · AnonReviewer1 · 2020-10-29
**A simple and effective domain adaptation reward modifier for transfer in RL**

**Rating:** 8
**Confidence:** 4

**Review:**

## Summary
This paper proposes a method for domain adaptation in RL where the source and target domains differ only in the transition distriubtions. A theoretical derivation based on RL as probabilistic inference is presented that starts with the objective of matching the desired distribution of trajectories in the target domain with the distribution achieved by the policy in the source domain. The final objective appears as a modification to the reward function while training in the source domain and is implemented easily with just two binary classifiers that predict the domain given either state-action or state-action-next-state tuples. Theorem 4.1 provides a theoretical guarantee on the performance of a policy trained on such a modified reward in the source domain by giving a bound on the performance in the target domain, under a very mild assumption that the optimal policy on the target domain achieves similar rewards when put in the source domain. Experiments are presented that show improved performance in terms of rewards vs experience on target domain on environments such as broken reacher, broken ant, etc (where the target domain has some "broken" component). Further, it is also shown that the reward modification on source visually matches the reward expected in target (Fig 4), that without the reward modification the policy usually exploits the source domain's transitions which cannot be exploited in the target domain, and finally, that safety emerges from the proposed objective.

## Strengths
- The theoretical bound presented in Theorem 4.1 is strong given that Assumption 1 is quite mild. In fact, Assumption 1 is trivially met in most of the environments used in Fig 6, e.g.: for half cheetah obstacle, the target policy does not run in to the obstacle and hence will get the same reward in the source and target environment.
- The final form of the objective as a modification to the reward is a simple and easy to implement objective with the two binary classifiers. Further, the max-entropy derivation makes it applicable to any maximum-entropy algorithm.
- The empirical evidence shows that the proposed method has very similar performance to the RL on target baselines while improving over other domain adaptation baselines in four continuous control environments.

## Weaknesses
- The variety in the source-target domain shift seems limited in the experiments shown in Figure 6. Three tasks have the crippled-limb setting and one task has an obstacle in the target domains. The introduction started off by giving some great examples such as aggressive driving failing to work on target domains with icy roads. It would have been great to see application of this method on more subtle but dangerous changes to the target domain, similar to the car driving example. For example, reducing friction on the HalfCheetah or Ant environments -- it would be quite interesting to see what strategies develop in the source domain; does the agent take shorter steps to staty in contact with the floor?
- The assumption that the rewards for the target and source domains match is suitable for the choice of environments in Section 6 but seems quite limiting to a more general setting where certain rewards may never be observed in the source domain. Further, there is an implicit assumption that the source domain is "free" and the target domain is "restrictive" (e.g. broken ant, obstacle halfcheetah). It would be good to discuss the rare but still possible case where the opposite is true, the target domain allows for more "free" movement.

## Other issues/comments
- Theorem 4.1 has a crucial typo: $p_{\text{source}}$ is mentioned on the right hand side but it should actually be $p_{\text{target}}$. The appendix (Theorem B.3) has the correct version of this theorem statement.
- In Theorem 4.1, the reward-maximizing (entropy-regularized) policy in the target domain is said to satisfy Assumption 1. To confim, does this mean that, in the target domain, the policy that maximizes the entropy-regularized reward objective is the same as the policy that maximizes rewards without the maximum-entropy constraint?
- Minor typo in Section 5, Algorithm Summary: $q_{\theta_{SAS}}$ should be $q_{\theta_{SA}}$ in the second line.
- I don't understand the transparent colored lines in Figure 6, are they supposed to represent error bars or are they the non-smooth version of the darker colored lines?
- Typo/stray sentence in the last para, last line of Section 1.

## Feedback to authors
- To further study the usefulness of this approach (as future work perhaps), it would be good to apply it to an offline RL setting where a finite set of "good" trajectories from the target domain are provided and no further trajectories can be collected during training. Only during test time is the agent then deployed in the target domain to measure performance (and possibly safety).


--------------------------
## Post-rebuttal Update

The authors have shown new experiments on icy environments that show good results for the proposed method (DARC). This directly addresses my point (and recommendation) about trying out experiments similar to the aggressive driving on icy road example that was mentioned in the introduction. Having read through the other reviews and responses by the authors, I feel that most major concerns have been addressed. As such I am inclined to increase my score from 7 -> 8, recommending acceptance of the paper and entrusting the authors to include the new experiments in the main paper.

A minor note: My second point under "Other issues/comments" section was not answered in the rebuttal. I hope the authors can clarify this in the future either in the main paper or appendix.

---

> ### Author Response · Authors · 2020-11-20
> **Response to R1**
>
> We thank the reviewer for the detailed review and for the suggestions for improvement. As suggested in the review, we ran an additional experiment on the HalfCheetah and Ant (and Walker) environments with varying friction. DARC successfully learns policies that perform well in environments with different friction. As visualized on the website (https://darc-anonymous.github.io/), the policies learned by DARC do learn to take shorter steps. We believe that this additional experiment and the clarifications to assumptions and writing (discussed below) address the main concerns of the review (please let us know if we missed a concern). We kindly ask the reviewer to revisit the review in light of these revisions.
> * **Writing**: We have revised the paper to address the typos mentioned.  We have also added a discussion of the assumption in Eq. 1, describing what would happen if the opposite held. This discussion is in Section 3 after Eq. 1.
> * **Assumption 1**: The interpretation of Assumption 1 is not quite correct. Rather, Assumption 1 says that the optimal policy for the target domain doesn't exploit the source domain. This reflects the intuition you noted before about the source domain being more "free" than the target domain, so a policy that works well in the target domain will also work well in the source domain. Note that we do not require the opposite to be true.
> * **Transparent lines in Fig 6**: Each transparent line corresponds to a different random seed, with darker lines corresponding to the average of these random seeds. We have updated the text to describe this.

---

### Author Response · Authors · 2020-11-20
**General Comment: New experiments on "icy" environments and discussion of assumptions**

We thank all the reviewers for their comments!

**New results on "icy" environments**: As suggested by R1 and R4, we ran additional experiments comparing the performance of DARC on locomotion tasks where slippery ice is included in the target domain, but not the target domain. DARC practices in the source domain (no ice) but learns policies that perform well in the target (icy) domain. A baseline that just performs RL on the source domain (no ice) performs poorly on the target (icy) domain. As suggested by R3, we also include an experiment where we vary the dynamics discrepancy. Please see this anonymous website for videos and plots: https://darc-anonymous.github.io/

**Assumptions**: We want to address one common concern about the assumptions about the source and target domain. Our analysis assumes that all outcomes possible in the target domain must be possible (perhaps with small probability) in the source domain. Said in other words, the agent cannot be "surprised" when placed in the target domain. This assumption is stated explicitly in Eq. 1; R1 found this assumption to be "very mild." While this assumption is needed for our theoretical guarantees, our results show that our method works in environments that do not satisfy this assumption (thanks to R2 for pointing this out!).

Without, it seems impossible to get any _theoretical_ guarantees. For example, if there is a wall in the source domain but there's a hole in the wall in the target domain, how could an agent practicing in the source domain possibly know about the existence or location of the hole in the wall? We have added this discussion in Section 3 after Eq. 1. Note that the most similar prior work [Wulfmeier '17] does not discuss this assumption, because this prior work does not aim to provide theoretical guarantees. In contrast, we show in Theorem 4.1 that our method learns a near-optimal policy for the target domain, despite practicing only in the source domain. Our theoretical result relies on this assumption; without this assumption, no guarantee is possible.

---

### Decision · Program_Chairs · 2021-01-07
**Final Decision**

**Decision:**

Accept (Poster)

**Comment:**

This paper presents an approach to domain adaptation in reinforcement learning. The main idea behind this approach, DARC, is to modify the reward function in the source domain so that the learned policy is optimal in the target domain. This is achieved by learning a classifier that learns to discriminate between the data from the source domain and those from the target domain.

Overall, reviewers appreciated the intuitiveness of the approach as well as its formal analysis. They had some concerns with respect to experiments, which was sorted out in the author response period. Given the overall positive reviews, I recommend accepting the paper.